# Prediction of Bead Geometry with Changing Welding Speed Using Artificial Neural Network

**DOI:** 10.3390/ma14061494

**Published:** 2021-03-18

**Authors:** Ran Li, Manshu Dong, Hongming Gao

**Affiliations:** 1State Key Laboratory of Advanced Welding and Joining, Harbin Institute of Technology, West Straight Street 92, Harbin 150001, China; 17b909110@stu.hit.edu.cn; 2Ningxia Tiandi Benniu Industrial Group Co., Ltd., Shizuishan 753000, China; dongmanshu@nxtdbn.com

**Keywords:** bead geometry, prediction model, welding parameter, artificial neural network

## Abstract

Bead size and shape are important considerations for industry design and quality detection. It is hard to deduce an appropriate mathematical model for predicting the bead geometry in a continually changing welding process due to the complex interrelationship between different welding parameters and the actual bead. In this paper, an artificial neural network model for predicting the bead geometry with changing welding speed was developed. The experiment was performed by a welding robot in gas metal arc welding process. The welding speed was stochastically changed during the welding process. By transient response tests, it was indicated that the changing welding speed had a spatial influence on bead geometry, which ranged from 10 mm backward to 22 mm forward with certain welding parameters. For this study, the input parameters of model were the spatial welding speed sequence, and the output parameters were bead width and reinforcement. The bead geometry was recognized by polynomial fitting of the profile coordinates, as measured by a structured laser light sensor. The results showed that the model with the structure of 33-6-2 had achieved high accuracy in both the training dataset and test dataset, which were 99% and 96%, respectively.

## 1. Introduction

Welding is an essential process in most industrial manufacturing. It is critical to realize automation in welding manufacturing due to the hard work condition and shortage of skilled workers. However, most of the welding robots applied in the automatic manufacturing now are still primary teaching-playback robots and off-line programming robots, whose welding parameters are set in advance and have a lack of adaptability to different work-piece with unpredictable manufacturing error and assembly error, and flexibility to irregular bead geometry with specific requests in actual work conditions [1,2]. Improving the manufacturing quality and assembly quality heavily raises the production cost, and the traditional “trial and error” method needs plenty of man hours. As such, an acceptable solution is the introduction of intelligent welding technology to improve the reliability of product and production efficiency [3,4]. For a given task, the intelligent welding system should eliminate the trial time substantially to specify welding parameters. Building a credible model for bead geometry prediction is an effective method for actual manufacturing.

Welding is a rather complex process with many crucial parameters, such as voltage, current, welding speed, groove geometry, etc. Because to the complex interrelationship between these welding parameters, it is hard to deduce an appropriate physics model in continuous welding process with changing parameters [5]. In recent decades, researchers have applied various mathematical models to build the relationship between multi-input and multi-output parameters, e.g., factorial design, linear and nonlinear regression, response surface methodology, and artificial neural network (ANN) [4,6,7,8,9,10,11]. These design of experiments (DOE) techniques apply to different areas according to the complex relationship between input and output parameters, and they achieve high accuracy and efficiency in modeling. However, in the research of bead geometry prediction, most researchers predict the bead geometry with constant parameters in these models. They change the welding parameters bead-by-bead and they do not change the parameters in one same bead. This may be insufficient for the actual welding process with changing groove and irregular desired geometry. Developing a dynamic model with changing parameters for welding process is still under research.

In recent years, ANN has been demonstrated to be a powerful tool in developing models for the interrelationships between various inputs and outputs in different areas, such as finance, medicine, and engineering. The universal approximation theorem states that a feed forward network with a single hidden layer containing a finite number of neurons can approximate any continuous functions on compact subsets of Rn [12]. Many researchers in welding area applied ANN in building model for parameters prediction in different welding applications. Nagesh et al. [13] built a back propagation neural network (BPNN) model with the inputs (electrode feed rate, arc-power, arc-voltage, arc-current, arc-length, and arc-travel rate), outputs (bead height, bead width, depth of penetration, and area of penetration) and yielded accurate results in shielded metal-arc welding. Shim et al. [14] used BPNN and response surface methodology to predict bead reinforcement area by welding voltage, arc current, welding speed, contact tube weld distance, and welding angle, and obtained good quality predictions in automatic gas metal arc welding. Kshirsagar et al. [15] used a two-stage algorithm that consisted of support vector machine (SVM) and an ANN to improve the prediction performance in automated tungsten inert gas (TIG) welding. Ding et al. [16] designed a BPNN model with three inputs (wire feed rate, travel speed, and stick-out), two outputs (bead height, bead width), and one hidden layer for shape prediction used in arc-welding-based additive welding. Ahmed et al. [17] compared the multilayer perceptron neural network (MLP-NN) and radial basis function neural network (RBF-NN) for predicting the bead shape by welding parameters in shielded metal arc welding, and found that RBF-NN was able to achieve a higher level of accuracy. Las-Casas et al. [18] used ANN to predict ferrite quantity and bead geometry with the inputs of welding voltage, current, different filler material, and obtained a small error percentage. 

The ANN has been proven its reliability to these researchers in predicting bead geometry and other properties with a rather small dataset, which is usually less than 100 samples, in almost all of the welding methods. Additionally, the ANN also has the advantages of continuous updating with new data, handling large number of inputs and outputs neurons, and filtering noises, showing its great potential in industrial manufacturing application [19]. While all researches mentioned above focus on the bead property prediction with stable welding parameters in one sample, which only meets laboratory requirement, not the actual manufacturing requirement. 

The bead size and shape are important considerations for industry design and quality detection. The most economic and reliable bead geometry depends on the actual desire and real weldment. Identifying the specific welding parameters for desired bead geometry is a hard work though costly and time-consuming trials. This makes it necessary to develop a more efficient method to specify the welding parameters. The common welding parameters include voltage, current, feed speed, and welding speed. The voltage, current, and feed speed are usually unified by advanced welding machine in automated manufacturing process to keep stable metal transfer. Changing the welding speed and path is a more accessible method to control bead profile.

In this paper, a three-layer back propagation neural network model is developed for predicting the bead width and reinforcement with stochastically changing welding speed in gas metal arc welding (GMAW) process. The experiments are carried out on plates. The welding speed is the only changing parameter in the study and the other welding parameters are constant. The ANN model is trained and tested with 465 samples. The sound result will provide the welding engineers with a new method to specify welding parameters with a special desired weld. In addition, all of the experiments are carried out on large-sized plate, and both ends of experimental beads are not adopted for sample acquisition, aiming to eliminate the influence of heat dissipation and accumulation in the welding process.

## 2. Experimental Details

### 2.1. Experimental System

The experimental system consists of a six-axis industrial robot, a structured light laser sensor, a welding machine, and a computer (Figure 1). All of these components are connected via Ethernet. The computer sends the movement command to the robot and the arc command to the welding machine. The sensor system extracts the laser stripe projected on the work-piece and then sends a 1024 × 2 coordinates array to PC, which indicates the geometry of the bead. The measurement resolution of the laser sensor system is 0.05 mm.

### 2.2. Materials and Welding Parameters

Q235 steel is the base material used for this study. The plate is cut into 300 × 150 × 10 mm^3^ pieces and then brushed to eliminate dirt and oxides. The wire material is ER70s-6 with a diameter of 1.2 mm. Table 1 shows the nominal chemical composition of Q235 steel and ER70s-6. The shielding gas is composed of 82% Argon and 18% CO_2_, with a flow rate of 15 L/min. The wire feed speed is set as 6 m/min. Additionally, the welding voltage is 21.4 V and welding current is 107 A. The experiments of constant welding speed show that, when the welding speed is in the range from 10 cm/min. to 60 cm/min. with other welding parameters above, a stable and uniform bead can be obtained. Additionally, the range of changing welding speed is limited from 10 cm/min. to 60 cm/min. in this study.

### 2.3. Bead Geometry Acquisition

Bead width and reinforcement are two key measures of bead geometry, which can be detected by welders directly as the main judge criteria of bead quality. Figure 2 shows these measures. 

The structured laser light sensor is widely used in the welding industry for seam tracking and bead detection [20]. By projecting a laser line onto the surface of work piece, the sensor obtains the 2D bead profile by CCD camera and then transforms the image to relative two-dimensional coordinates. Subsequently, the whole bead profile can be obtained as the laser sensor moved along with the welding torch. Extracting the bead feature values is still a difficult task. The traditional methods to extract the turning points are template matching and gradient calculation [21,22]. In this paper, a polynomial function is introduced to extract the turning points.

In most of the welding conditions, the base plate is usually flat, which should perform a straight line in sensor vision. However, the extracted laser line is usually slightly curved and angled in practice, due to the uneven surface or heat distortion. A polynomial function is used to model the base plate surface by the least square method (LSM). The polynomial function can be expressed as,
(1)$$y_{i}=a_{0}+a_{1}x_{i}+a_{2}x_{i}^{2}+\cdots +a_{n}x_{i}^{n}$$
where *x_i_*, *y_i_* are the 2D coordinates of the points on the laser line. 

The algorithm first finds the peak point of the profile as the center of the bead. Subsequently, the regions of left surface and right surface are segmented by pre-set value, which equals the max bead width. The point sets in two surface regions are used to optimize Equation (1) by the LSM. The fitting results (Figure 3) show that quadratic fitting is suitable for plate welding conditions, while it conforms to the distortion of plate surface properly and it is robust to small dataset noise. By calculating the deviation in *Y*-axis to the fitting curves in the middle region, the turning points can be found when the deviations of continuous points that are above threshold value are greater than the average deviations in two surface regions. The top point of bead is found by calculating the distance between the points among turning points and the base plate surface. Subsequently, the bead width and reinforcement are then calculated according these feature points. The time processing these calculations is 15.9 ms on average. Additionally, the average error is 0.076 mm by detecting a standard V-groove with precision machining.

## 3. Model Development

### 3.1. Transient Response Tests

In order to define the input layer range and build the mapping relationship between welding speed and bead geometry, transient response tests are carried out with minimum welding speed, 10 cm/min., and maximum welding speed, 60 cm/min, which is tested in constant parameter welding tests. Figure 4 and Figure 5 show the results. When welding speed steps from 10 cm/min to 60 cm/min, the bead width keeps decreasing in the range from 2 mm backward to 14 mm forward, and it increases to a stable value at 20 mm forward, the bead reinforcement keeps decreasing in the range from 9 mm backward to 13 mm forward, and it increases to a stable value at 22 mm forward. When welding speed steps from 60 cm/min to 10 cm/min, the bead width keeps increasing in the range from 6 mm backward to 16 mm forward, the bead reinforcement keeps increasing in the range from 10 mm backward to 12 mm forward. The first bead shows a shrinkage 10 mm behind the step point, while the second bead changes more smoothly. This phenomenon is mainly caused by the flow of weld pool and the movement of heat source. Finally, the inputs neurons are defined with spatial welding speed sequence from 10 mm backward to 22 mm forward, totaling 33 neurons.

### 3.2. Model Development

BPN is widely used in mathematical model research for its non-linear mapping between the input and output parameters. The neurons are full connected between two adjacent layers. The weights are set randomly in advance. The activation function transforms values to a scaled output value in a higher level. The error is estimated as the difference between the actual and computed outputs. This procedure constitutes forward flow of back propagation phase and error computed is back propagated through same network to update weights. Weight change is calculated for all connections. The errors for all patterns are summed and the algorithm is active until the error falls below a specified value [10,11,13].

In this study, a basic three-layer back propagation neuron network is established. The input layer is the spatial welding speed sequence, including 33 neurons. Additionally, the output layer is the bead geometry values, including bead width and bead reinforcement. The size of hidden layer is one of the most important considerations for the ANN model. Chiu et al. [23] found that the increase in hidden neurons did slightly increase the accuracy in part, but not all datasets. To avoid overfitting and under fitting, the number of hidden neurons is tested from 2 to 18 in step 2. Subsequently, a typical three-layer back propagation is developed according to this task, as shown in Figure 6.

### 3.3. Datasets Acquisition

It is critical to achieve a large enough dataset for ANN model training and test to correct the network parameters and validate the model performance of practical accuracy. In addition to transient response tests, three other beads of 200 mm in length undergo the GMAW process. The welding speeds are stable in both ends of 50 mm, 10 cm/min., or 60 cm/min., and steps to a random value per 10 mm in the middle 100 mm. From each bead, 133 samples can be extracted by the experimental system, including the spatial welding speed sequence, bead width, and bead reinforcement. Subsequently, a training dataset of 332 samples and a test dataset of 133 samples are obtain. 

### 3.4. Training Process

The development and the training of the network are carried out on a PC using PyTorch framework on Python. The learning rate is 0.01, the batch size is 32, the epochs are 30, the activation function is Tanh, and the other parameters are all defaults.

Figure 7 shows the training. In the first 50 steps, the training MSE decreases linearly and it is volatile. In step 50 to step 100, it decreases slowly to a stable value. The test MSE decreases with the training MSE accordingly but stably. From step 150 to the end, they both stay at a stable value, which is about 0.0018.

### 3.5. Results and Discussion

Table 2 shows the training and test results of different number of hidden neurons. The lowest training MSE is achieved in the 33-10-2 structure, while the highest training MSE is achieved in the 33-2-2 model. Additionally, the lowest test MSE is achieved in the 33-6-2 model, while the highest MSE is obtained in the 33-4-2 model. No further reduction of training MSE or test MSE is achieved when the number of hidden neurons increases. The test MSEs are all slightly higher than the training MSEs in all models. The 33-6-2 model structure is adopted for further development because the increase of the number of hidden neurons will decrease the generalization ability of the ANN model.

Figure 8 shows the comparison of the prediction results from the ANN model and actual values in test dataset. The results show little deviation between the predicted value and actual value. The highest error is 0.899 mm and the average error is 0.23 mm in bead width. Additionally, the highest error is 0.534 mm and the average error is 0.09 mm in bead reinforcement. On one hand, these errors are caused by the lack of a large number of training samples, which may influence the generalization ability of the ANN model. On the other hand, the welding process is not stable compared with other process technology. Because to the influence of the fluctuations from droplet transition and spatter. The bead width and reinforcement deviation are unavoidable, which introduces inevitable system error. However, this precision is relatively acceptable in the welding process. 

The prediction results in the test dataset show that the ANN model can accurately predict bead width and reinforcement. Additionally, no counterintuitive data with unacceptable deviation are detected in this work.

## 4. Conclusions

In this study, an ANN model is developed to predict bead geometry in a continuously changing welding condition, in which the welding speed changes stochastically. Additionally, the following conclusions have been made.

The welding parameter at one position has lasting infection on bead formation in a long range along welding direction, which has been proved by transient response tests. Building a mathematical model in dynamic welding process should involve the spatial relationship of the changing welding parameters in a large enough range. 

A basic three-layer back propagation neuron network shows a high accuracy in predicting bead geometry with a spatial welding speed sequence, which changes stochastically while other welding parameters keep constant. The testing results show little deviation between the actual bead geometry values and the predicted bead geometry values. This makes it practicable to introduce the ANN model in the actual changing welding process.

This work only changed the welding speed on plates with no groove. Future work will focus on the prediction model of bead geometry with more input parameters. Not only welding speed, but also groove geometry, welding path, and weaving parameters will be brought into the prediction model, which will meet the actual welding demand. Finally, a reinforcement learning model will be established to control bead formation for an uneven workpiece based on this prediction model.

## Figures and Tables

**Figure 1 materials-14-01494-f001:**
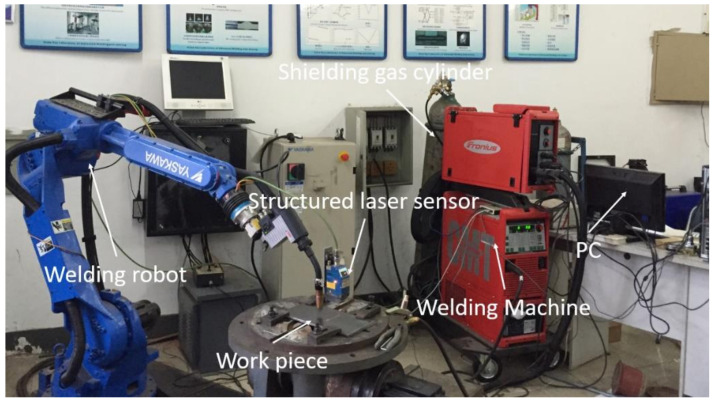
Experiment system.

**Figure 2 materials-14-01494-f002:**
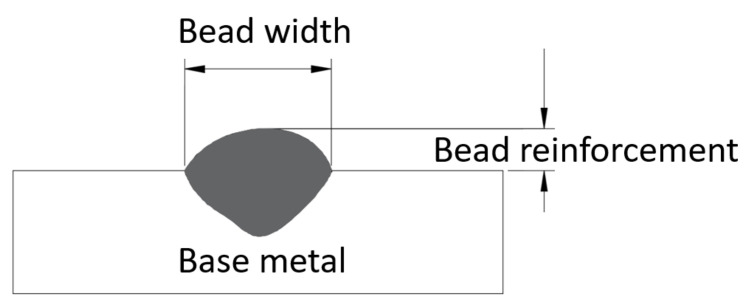
Weld bead geometry.

**Figure 3 materials-14-01494-f003:**
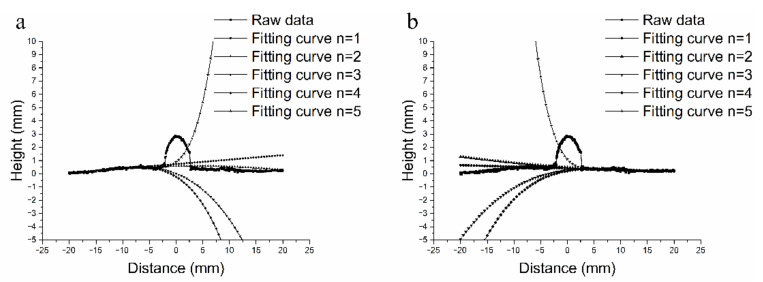
Fitting results. (**a**): left surface. (**b**): right surface.

**Figure 4 materials-14-01494-f004:**
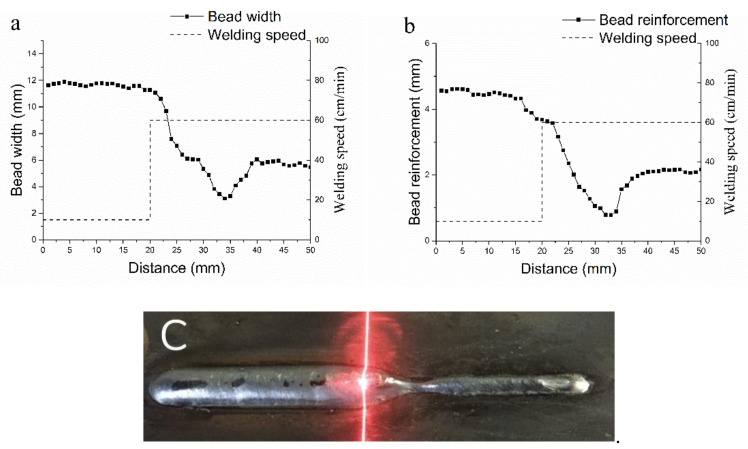
Results of transient response test (welding speed steps from 60 cm/min to 10 cm/min.). (**a**): bead width value; (**b**): bead reinforcement value; and, (**c**): actual bead.

**Figure 5 materials-14-01494-f005:**
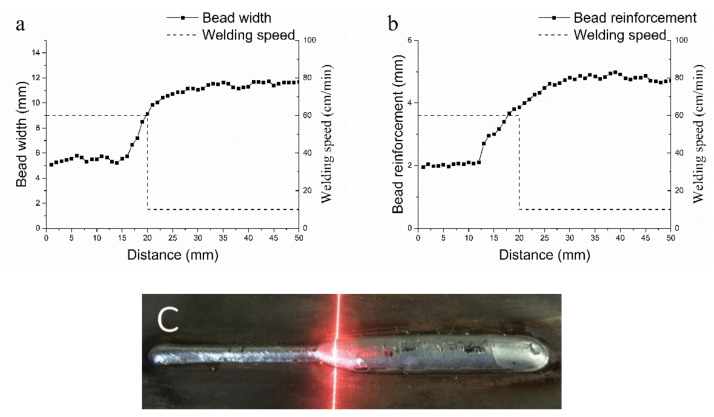
Results of transient response test (welding speed steps from 60 cm/min to 10 cm/min.). (**a**): bead width value; (**b**): bead reinforcement value; and, (**c**): actual bead.

**Figure 6 materials-14-01494-f006:**
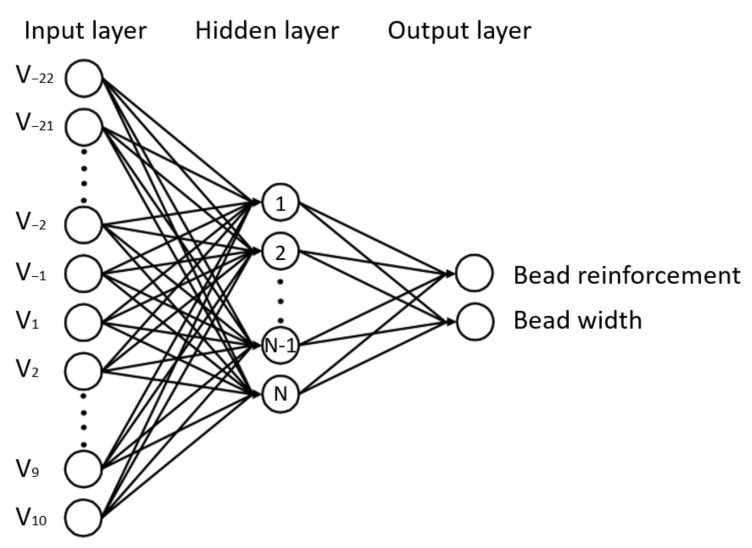
Artificial neural network (ANN) model for bead geometry prediction.

**Figure 7 materials-14-01494-f007:**
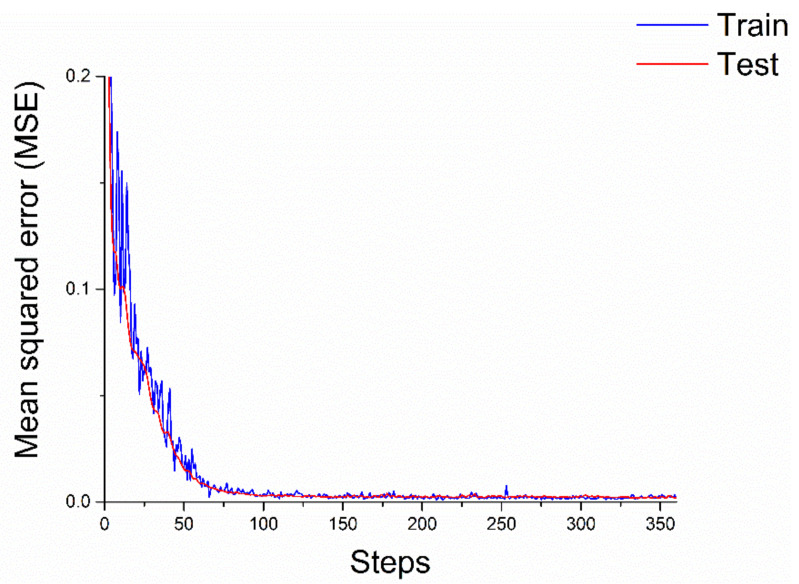
The training and test in process.

**Figure 8 materials-14-01494-f008:**
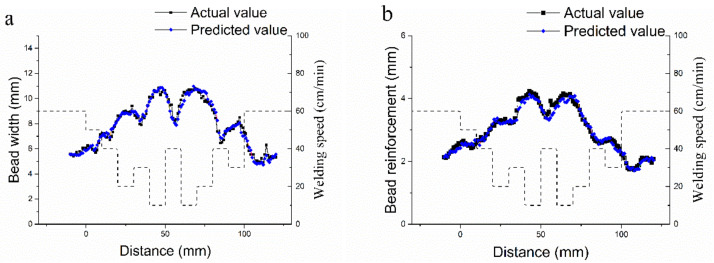
Comparison of predicted and actual results in test dataset: (**a**): bead width value; and, (**b**): bead reinforcement value.

**Table 1 materials-14-01494-t001:** Nominal chemical composition of Q235 steel and ER70s-6 (wt.%).

Material	C	Mn	Si	S	P	Cr	Ni	Cu	Fe
Q235	0.17 max	0.35–0.80	0.30 max	0.035 max	0.035 max	0.03 max	0.03 max	0.3 max	Bal.
ER70s-6	0.06–0.15	1.40–1.85	0.80–1.15	0.04 max	0.03 max	0.15 max	0.15 max	0.5 max	Bal.

**Table 2 materials-14-01494-t002:** Training and test results of different hidden neurons.

Structure	Training MSE	Test MSE
33-2-2	6.7624 × 10^−3^	7.8341 × 10^−3^
33-4-2	2.5074 × 10^−3^	7.3037 × 10^−3^
33-6-2	1.9432 × 10^−3^	6.372 × 10^−3^
33-8-2	2.1679 × 10^−3^	8.3761 × 10^−3^
33-10-2	1.3076 × 10^−3^	7.8045 × 10^−3^
33-12-2	3.0185 × 10^−3^	8.2483 × 10^−3^
33-14-2	3.5779 × 10^−3^	6.888 × 10^−3^
33-16-2	2.3568 × 10^−3^	6.5465 × 10^−3^
33-18-2	4.5394 × 10^−3^	7.3875 × 10^−3^

## Data Availability

Data sharing is not applicable to this article.

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
