# Peer review of "Prediction of Bead Geometry with Changing Welding Speed Using Artificial Neural Network"

_materials, 2021, doi:10.3390/ma14061494_

Round 1

Reviewer 1 Report

The paper deals with the prediction of bead geometry in gas metal arc welding. The parameters considered for the study are welding speed, bead width, and reinforcement. The Artificial Neural Network (ANN) technique is used to make predictions on variations. The paper is generally well written, and the area is worthy of investigation with potential impact on the evolution of artificial intelligence in the field. The reviewer recommends the following revisions to improve the clarity of the manuscript.

  1. Clarify the term bead reinforcement.
  2. What software was used to develop ANN. Detailed information is required so that the study can be recreated.
  3. Please justify the need for this study and clarify its novelty in the introduction section.
  4. I also recommend widening the literature review and compare how the technique is different from DoE based prediction such as the ones shown in:

Additively manufactured AlSi10Mg inherently stable thin and thick-walled lattice with negative Poisson’s ratio, Composite Structures Volume 247, 1 September 2020, 112469.

A Virtual Design of Experiments Method to Evaluate the Effect of Design and Welding Parameters on Weld Quality in Aerospace Applications. Aerospace 2019, 6, 74. https://doi.org/10.3390/aerospace6060074

  1. How are the weld bead dimensions characterised?
  2. It is not clear how the neural network model and training are carried out for the results presented. Can I please urge the authors to present this in detail so that fellow researchers can benefit from the methodology?
  3. How was validation performed for results obtained? Please clarify?
  4. It is important to report any counterintuitive data obtained, this is a common scenario when using ANN networks.
  5. Please present a brief description of the future direction of the research.

Author Response

1. The bead reinforcement is also called bead height. It means the height of deposited metal above base metal. The three key geometry parameters of weld bead are bead width, bead reinforcement and bead penetration. The bead width and reinforcement can be detected directly by laser sensor per millimeter. They are also the main criteria for welders to judge the bead quality in welding process. While the bead penetration is not available per millimeter because of the dissymmetry of bead geometry. And it cannot be detected by welders in actual welding process. The bead width and bead reinforcement are illustrated in added Fig.2 in section 2.3.

The dissymmetry of bead geometry is shown in the article:

Weld parameter prediction using artificial neural network: FN and geometric parameter prediction of austenitic stainless steel welds.

https://doi.org/10.1007/s40430-017-0928-0

2. The development and the training of the network are carried out on a PC using PyTorch framework on Python. The learning rate is 0.01, the batch size is 32, the epochs are 30, the activation function is Tanh and the other parameters are all defaults. The detailed information is added in paragraph 1 in section 3.4

3. The introduction section is re-written.

Paragraph 1-2 shows the need for this study and its novelty. Firstly, intelligent welding technology is urgently needed to improve the reliability of product and production efficiency. Secondly, the bead geometry prediction model can decrease the “trial and error” time. Thirdly, most researchers now build the ANN model with stable parameters in one bead. In this paper, we build the prediction model with changing welding speed which meets the actual manufacturing requirement. And in paragraph 3-4 we show the research status in details.

4. The literature review is improved according to this advice. 

5. The bead geometry is detected by active laser sensor. By projecting a laser line onto the surface of work piece, the sensor gets the 2D bead profile by CCD camera and transforms the image to relative two-dimensional coordinates. A polynomial function is used to model the base plate surface by the least square method on two sides. Then the deviation between actual value and function value is calculated in Y-axis. When several continuous points with deviation above threshold value are detected, the turning points are detected. By calculating the distance between the points among turning points and the base plate surface, the top point of bead is found. Then the bead width and reinforcement are then calculated according these feature points. This is shown in section 2.3, and this section is also re-written.

6. The procedure is shown in section 3. At first, we explored the influence range of welding speed, by which we defined the input neuron range and the mapping relation between welding sequence and bead measures. Secondly, we chose an appropriate ANN model structure for this work. Thirdly, we collected data from new beads with varying welding speed, which changed stochastically, for building training dataset and test dataset. Fourthly, we trained the ANN model with the training dataset until the training MSE reached a stable value. Finally, we calculated the errors between predicted bead measures from the trained model and actual values in the test dataset to validate the performance of the ANN model.

7. In this study, 5 beads are welded for samples acquisition, 4 beads for training dataset, including 2 transient response test beads and 2 long beads, and 1 long bead for test dataset. The welding speed changes stochastically in long beads. The training MSE is 0.0018.

In test dataset, the highest error is 0.899 mm and the average error is 0.23 mm in bead width. And the highest error is 0.534 mm and the average error is 0.09 mm in bead reinforcement. This precision is relatively acceptable in welding process.

8. The welding process is not stable compared with other process technology. Due to the influence of droplet transition and spatter. The bead width and reinforcement deviation are unavoidable. In a long bead on plate welded recently with all welding parameters keep stable, the max deviation, average deviation and average value of bead width are 0.723mm, 0.25mm and 10.21mm respectively; the max deviation, average deviation and average value of bead reinforcement are 0.145mm, 0.046mm and 2.83mm respectively. In this study, the detected data are all accepted in datasets. The counterintuitive data should be also accepted as system noise. Once However, no counterintuitive data with unacceptable deviation is detected in this work.

9. The future work is shown at the end of the article. This study only changes welding speed on plate at present. In the future, more changing welding parameters will be added in the model, including groove geometry, welding path, weaving parameters, etc. However, too many input neurons will cause the curse of dimensionality and need a great number of welding tests for datasets. The research direction will focus on dimensionality reduction and framework optimization.

Reviewer 2 Report

The manuscript “Prediction of bead geometry with changing welding speed using artificial neural network” is an interesting research work. The manuscript is well-written with just some minor issues. It is also well-structured, beginning with a nice introduction, and clearly identifying the work objectives. The methodology is clearly described and presented, and so are the results.

It aims to reduce trial and error optimization, not just regarding welding experiments, but also considering bead geometry changes instead of constant geometries as reported in some numerical studies in the literature. Nevertheless, the introduction needs significant improvements. First, the authors need to improve the literature review. Several important articles were not addressed. Some suggestions:

  • https://doi.org/10.1016/S0924-0136(02)00101-2
  • https://doi.org/10.1177/1687814018781492
  • https://doi.org/10.1016/j.promfg.2017.09.052
  • https://doi.org/10.3390/jmmp3020039

Second, and facing the state of the art, authors need to clearly identify the innovative aspects of this work in comparison with the literature. Other issues are:

  • How many samples were tested?
  • What were the precisions in the measuring of bead geometry? Please provide this information with some error info, e.g. graphical data with error bars~
  • What is the total time necessary to process these calculations?
  • Please comment on the possibilities on using this model for other cases base on its validation.
  • Graphical data (plots) have overlapped information
  • Please substitute the figures of the beads by some with better quality. It is not possible to properly assess it
  • Some minor issues: “mm3”; “CO2”,

Author Response

1. The literature review is improved according to this advice. 

2. The introduction section is re-written.

Paragraph 1-2 shows the need for this study and its novelty. Firstly, intelligent welding technology is urgently needed to improve the reliability of product and production efficiency. Secondly, the bead geometry prediction model can decrease the “trial and error” time. Thirdly, most researchers now build the ANN model with stable parameters in one bead. In this paper, we build the prediction model with changing welding speed which meets the actual manufacturing requirement. And in paragraph 3-4 we show the research status in details.

3. The training dataset includes 332 samples and the test dataset includes 133 samples.

4. In this study, a META 50V1 Smart Laser System is used for bead detection. By projecting a laser line onto the surface of work piece, the sensor gets the 2D bead profile by CCD camera and transforms the image to relative two-dimensional coordinates. The resolution of this sensor system is 0.05mm. A polynomial function is used to model the base plate surface by the least square method on two sides. Then the deviation between actual value and function value is calculated in Y-axis. When several continuous points with deviation above threshold value are detected, the turning points are detected. By calculating the distance between the points among turning points and the base plate surface, the top point of bead is found. Then the bead width and reinforcement are then calculated according these feature points. A standard V-groove is experimented, which is 10mm wide and 5mm deep after precision machining. The average error is 0.076mm. This is shown in section 2.3, and this section is also re-written.

5. The total time processing these calculations is 15.9 ms in average. The maximum processing time is 17.2 ms and the minimum processing time is 14.5 ms in test.

6. In this study, we build an ANN model for bead width and reinforcement prediction, which shows high accuracy in validation. It provides a credible model for welding parameter programming with desired bead geometry.

This study only changes welding speed on plate at present. In the future, more changing welding parameters will be added in the model, including groove geometry, welding path, weaving parameters, etc. However, too many input neurons will cause the curse of dimensionality and need a great number of welding tests for datasets. The research direction will focus on dimensionality reduction and framework optimization.

7. These plots with overlapped information are corrected.

8. The figures are substituted.

9. These errors are checked and corrected.

Reviewer 3 Report

I think the research topic is very important.
Predicting the size of welding beads will be in demand not only in welding but also in additive manufacturing (VAAM).
Unfortunately, only one factor varied. It would be better if there were several listed authors (groove geometry, welding path, weaving parameters, etc.).
The heat dissipation conditions are very important. Even with constant technological parameters (welding speed, welding power, etc.), the width of the bead at the beginning and at the end of the trajectory when surfacing on a small plate will be different (the bead will expand as the plate warms up).
The given model can be adjusted only when welding large-sized parts without preheating. I propose to reflect these points in the article

I suggest improving:

It is necessary to expand p. 3.5 Results. in particular, I did not understand by what law the speed varied in order to obtain the change in the roller geometry indicated in Fig. 7

Author Response

1. This study only changes welding speed on plate at present. It is an attempt to build a credible prediction model with changing welding parameters. While most researchers predict the bead geometry with constant parameters in their models as shown in paragraph 3 in section 1. In the future, more changing welding parameters will be added in the model, including groove geometry, welding path, weaving parameters, etc. However, too many input neurons will cause the curse of dimensionality and need a great number of welding tests for datasets. The research direction will focus on dimensionality reduction and framework optimization.

2. In this work, the beads are all welded on large-sized plates. And the both ends of beads are not adopted in datasets. This may cause misunderstanding. These points are added in the end of the section 1 in the article.

3. The results section is re-written.

4. The Fig.7 is remade with the addition of welding speed sequence. The welding speed changes stochastically to random value generated by computer.

Round 2

Reviewer 2 Report

All my questions  and requests/comments were properly answered and adressed. No further questions or comments

Reviewer 3 Report

I offer to accept the job. good luck to the authors